# Response of Winter Wheat (*Triticum aestivum* L.) to Fertilizers with Nitrogen-Transformation Inhibitors and Timing of Their Application under Field Conditions

Marie Školníková [1], Petr Škarpa [1], Pavel Ryant [1], Zdenka Kozáková [2] and Jiří Antošovský [1,*]

1   Department of Agrochemistry, Soil Science, Microbiology and Plant Nutrition, Faculty of AgriScience, Mendel University in Brno, Zemědělská 1, 61300 Brno, Czech Republic; mar.skolnikova@seznam.cz (M.Š.); petr.skarpa@mendelu.cz (P.Š.); pavel.ryant@mendelu.cz (P.R.)
2   Faculty of Chemistry, Brno University of Technology, Purkyňova 118, 61200 Brno, Czech Republic; kozakova@fch.vutbr.cz
*   Correspondence: jiri.antosovsky@mendelu.cz

**Abstract:** Winter wheat is a widely cultivated crop that requires high inputs of nitrogen (N) fertilization, which is often connected with N losses. The application of fertilizers with nitrification (NI) and urease inhibitors (UI) is an opportunity to eliminate the risk of N losses and improve N availability to plants. The aim of this study is to compare the effect of conventional nitrogen fertilizers with fertilizers containing nitrogen-transformation inhibitors as well as to evaluate the timing of their application on the wheat-grain yield and quality under the conditions of a three-year field experiment. The examined fertilizers with inhibitors were applied in a single dose or in a split application in combination with conventional fertilizers. The single application of urea with NI and/or UI resulted in a relatively average increase in the grain yield, while protein content and the Zeleny-test values were significantly increased compared to the split N application. The more significant effect of urea with NI and UI was found under the moisture-rich conditions compared to the drier conditions. A significant increase in the grain yield (by 6.3%) and in the Zeleny-test value (by 16.5%) was observed after inhibited urea application comparing to the control treatment (without inhibitors).

**Keywords:** wheat; nitrification and urease inhibitors; split and single application of fertilizer; grain yield; quality of grain

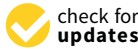



## 1. Introduction

The efficient use of fertilizers is an important factor of sustainable agriculture. It not only influences crop productivity, but it also reduces nutrients losses, which eliminates the detrimental impact on the environment. Nitrogen (N) is an essential nutrient for plant growth and development. Many authors have described its positive effect on the plant biomass production, the grain yield, and the grain protein content [1–3]. The average N efficiency in the field conditions is 32%, and no more than 40% [4]. The efficiency can reach $50-70\%$ in the case of a synchronic effect of N supply and crop demand [5]. Low efficiency of nitrogen fertilizers not only causes economical losses, but it is also environmentally unsafe (it represents leaching losses of $NO_3^-$, volatilization losses of $NH_3$, and emission of other N-containing gases) [6]. These losses decrease soil fertility, but also represent a risk to the atmosphere, hydrosphere and human health [7,8]. $NH_3$ volatilization is mainly involved in surface-applied fertilizers without incorporation into the soil [9]. N losses through $NH_3$ volatilization can reach up to 60 % of the nitrogen applied by the fertilization [10], and they are responsible for the generation of condensation nuclei, which contribute to the greenhouse effect [11,12].

Wheat is among the most globally cultivated cereal crops [13]. Under conditions of intensive agriculture systems, the traditional wheat cultivation requires high inputs

of N fertilizers which is related to the risk of N losses [14]. The nitrogen fertilization is considered crucial for an optimal crop yield. The split application of nitrogen is a common strategy for the optimization of the plants' nutrient uptake and for the reduction of the risk of N losses in conventional agriculture [15]. Positive effects of the split application on the wheat-grain yield, flour quality, and proportions of gliadin and glutenin have been described by several authors [16–18]. Another opportunity to reduce N losses and improve N efficiency is in the use of fertilizers with inhibitors. These effective and environmentally friendly fertilizers contain inhibitors that temporary restrict N changes in the soil (urea hydrolysis or nitrification). The most widely used N fertilizers in the world for the winter wheat production are ureic fertilizers. Urea that is applied to the soil surface is hydrolyzed very quickly, thereby generating $CO_2$ and high amounts of $NH_3$ [11]. The urea hydrolysis requires a urease enzyme, the activity of which could be reduced by a urease inhibitor (UI). The application of a UI with the urea helps to delay the conversion of urea into $NH_4^+$ in the soil [19,20] due to the partial inhibition of the urease activity [21]. The principle of most nitrification inhibitors (NIs) is the influence of the ammonia mono-oxygenase enzyme, which is responsible for the oxidation of ammonium into nitrite in the first step of nitrification (conversion of $NH_4^+$ into $NO_2$). NI temporarily binds ammonia mono-oxygenase, which leads to the conservation of immobile $NH_4^+$ in the soil for a longer period (4−10 weeks). Subsequently, it reduces the amount of $NO_3^-$, which is very mobile in the soil and is involved in leaching and denitrification [22,23]. Denitrification is also a source of NO and $N_2O$ emissions [24]. NIs are also recommended by The Intergovernmental Panel on Climate Change (IPCC) as an option to reduce $N_2O$ emissions in agriculture [25]. Stabilized fertilizers seem to represent a good opportunity to minimalize the negative impact of N losses on the environment and to improve the agronomic benefits of fertilization since their positive environmental aspects have been reviewed by many authors [4,26–30].

The amount of available N in the soil is influenced not only by fertilization but also by the level of mineralization. Mineralization of N is assumed to supply random and unpredictable amounts of inorganic N from one year to the next [31]. The fluctuation of residual soil N is unpredictable [32] and it is affected by random environmental effects [33]. The mineralization of organic soil N is often accelerated by the application of N fertilizers, which results in interactions of the added N or in a priming effect [34]. N supplied by fertilization can be conserved through its immobilization by micro-organisms (a biotic process) and fixation by soil-clay minerals (an abiotic process). Subsequently, it can be remineralized and further released in order to cover crop demands, thus reducing the N losses [35]. The application of N-transformation inhibitors significantly affects these processes [36].

Another essential nutrient for optimal wheat development is sulfur (S), which plays an important role in the yield formation and protein constitution [37,38]. The interaction between nitrogen and sulfur has an impact not only on the uptake and assimilation of $NO_3^-$ and $SO_4^{2-}$, but also on N and S metabolism [8]. Sulfur also positively influences the quality of protein in baking [39]. In wheat with S deficiency, asparagine amino acid is accumulated in the grain, which contributes to a higher risk of an unhealthy acrylamide formation during the baking of flour products [40].

This work should contribute to the description of the effect of nitrogen and sulfur fertilizers with inhibitors in combination with single and split applications and emphasize their effect on the winter wheat-grain yield and quality parameters. Based on previous research, the positive effect of NI and UI on the prolongation of nitrogen availability in the soil is expected. Therefore, the basic aim of this work is to confirm if the single or split application of nitrogen fertilizers with inhibitors provides similar or better results in terms of the grain yield and qualitative parameters of winter wheat in comparison with common technology (fertilization without inhibitors split into three doses).

## 2. Materials and Methods

### 2.1. Experimental Site and Field Treatments

The 4-year field experiment (2018–2021) was established as a small-plot field observation at the experimental station Žabčice in southern Moravia, the Czech Republic (49°1′18.658″ N and 16°36′56.003″ E). The region is characterized by warm and dry conditions; the annual precipitation ranges from 380−550 mm and the average annual temperature is 10.1 °C. The experiment was established on a silty clay loam soil (clay 38.0%; silt 46.3%; sand 15.7%); the soil type was stagnic fluvisols (FL-st). Each year, the experiment was based on a new plot within the experimental station. The basic physical–chemical parameters of the soil that were determined before the sowing are given in Table 1. The soil nutrient content characterizes the whole area used for the experiment in each growing season. The effect of the soil parameters, including N released by mineralization, was assumed to be the same throughout the experimental area.

**Table 1.** The physical–chemical properties of the soil.

| Soil Parameter | Growing Season | | | | | Refs. |
|---|---|---|---|---|---|---|
| | 2017/2018 | 2018/2019 | 2019/2020 | 2020/2021 | Average | |
| Cox (%) | 1.32 | 1.43 | 1.33 | 1.36 | 1.36 | [41] |
| CEC (mmol/kg) | 219 | 257 | 208 | 234 | 230 | [42] |
| pH/CaCl$_2$ | 6.8 | 6.6 | 6.4 | 5.9 | 6.4 | [42] |
| P (mg/kg) | 148 | 134 | 152 | 92 | 132 | [42] |
| K (mg/kg) | 276 | 247 | 283 | 184 | 248 | [42] |
| Ca (mg/kg) | 3644 | 3321 | 3641 | 3934 | 3635 | [42] |
| Mg (mg/kg) | 384 | 397 | 411 | 355 | 387 | [42] |
| SO$_4{}^{2-}$ (mg/kg) | 14.5 | 11.3 | 11.1 | 10.4 | 12 | [42] |
| NH$_4{}^+$ (mg/kg) | 1.84 | 2.69 | 1.49 | 9.73 | 4 | [42] |
| NO$_3{}^-$ (mg/kg) | 3.86 | 14.00 | 4.62 | 14.30 | 9 | [42] |

CEC—Cation exchange capacity; Cox—Soil oxidizable carbon.

The model crop used in this experiment was winter wheat of the Julie variety (Selgen, Prague, Czech Republic). This variety is characterized by a high yield potential and good disease resistance. It belongs to the quality class E, its stated protein content is 13.8%, the density of its grain is 80.4 kg/hL, and its Zeleny-test value is 60 mL [43].

Wheat crops were damaged by pests (voles) in the 2019/2020 growing season. The damage was so severe that it was impossible to assess the impact of the fertilizers' application. Therefore, the results of this season were not included in the evaluation of the 4-year experiment. The experiment was set up using a randomized complete-block design with nine treatments (Table 2); each treatment was repeated four times. The size of each experimental plot for the fertilization was 15 m$^2$. All fertilizers were spread by hand separately on each block. The examined fertilizers used in the experiments and their basic characteristics are listed in Table 3.

**Table 2.** Treatments and doses of fertilizers.

| Treatments | Term (T) of Fertilization (Dose of N, S kg/ha) | | | Total Dose of N, S (kg/ha) |
|---|---|---|---|---|
| | T1 (BBCH 25) | T2 (BBCH 32) | T3 (BBCH 50) | |
| control | CAN (55, 0) | CAN (65, 0) | UAN (40, 0) | 160, 0 |
| N1 | ALZON neo-N (160, 0) | | | 160, 0 |
| N2 | CAN (55, 0) | ALZON neo-N (105, 0) | | 160, 0 |
| N3 | UREA[stabil] (160, 0) | | | 160, 0 |
| N4 | CAN (55, 0) | UREA[stabil] (105, 0) | | 160, 0 |
| NS1 | CAN (55, 0) | ASN (105, 52) | | 160, 52 |
| NS2 | ASN (120, 60) | | UAN (40, 0) | 160, 60 |
| NS3 | ENSIN (160, 80) | | | 160, 80 |
| NS4 | ENSIN (120, 60) | | UAN (40, 0) | 160, 60 |

BBCH−phase of growing according to Lancashire [44], BBCH 25–tillering; BBCH 32−stem elongation, BBCH 50–beginning of heading. UAN—Urea ammonium nitrate; CAN—Calcium ammonium nitrate; ASN—Ammonium sulphate nitrate; ENSIN—Ammonium sulphate nitrate with NIs; UREA[stabil]—urea with UI; ALZON neo-N−urea with UI and NI.

**Table 3.** Type of used fertilizers.

| Fertilizers | Nutrients Content (%) | | Inhibitors | Producer |
|---|---|---|---|---|
| | N | S | | |
| ALZON neo-N | 46 | 0 | nitrification (NI) and urease (UI) | (SKW Piesteritz, Wittenberg, Germany) |
| UREA[stabil] | 46 | 0 | urease (UI) | (AGRA GROUP a.s., Střelské Hoštice, the Czech Republic) |
| ENSIN | 26 | 13 | nitrification (NI) | (Duslo, a.s., Šaľa, the Slovak Republic) |
| UAN | 30 | 0 | none | (ADW AGRO, a.s., Okříšky, the Czech Republic) |
| CAN | 27 | 0 | none | (Duslo, a.s., Šaľa, the Slovak Republic) |
| ASN | 26 | 13 | none | (Duslo, a.s., Šaľa, the Slovak Republic) |

UAN—Urea ammonium nitrate; CAN—Calcium ammonium nitrate; ASN—Ammonium sulphate nitrate; ENSIN—Ammonium sulphate nitrate with NIs; UREA[stabil]—urea with UI; ALZON neo-N—urea with UI and NI. ALZON neo-N contains NI: (MPA−N-[3(5)-methyl-1H-pyrazol-1-yl) methyl] acetamide) and UI: (2-NPT−N-(2-nitrophenyl) Phosphoric Triamide); UREA[stabil] contains UI: NBPT−N-(butyl) Thiophosphoric Triamide; ENSIN contains NIs: DCD—dicyandiamide, TZ—triazol.

Table 4 presents the terms of sowing, fertilizer applications and the date of the harvest. In all experimental years, *winter wheat* was grown after winter wheat (a pre-crop). Winter wheat, grown as the pre-crop, was cultivated identically throughout the experimental area in each year. It was fertilized with a N fertilizer at the same rate (120 kg/ha N). The harvest was performed at the stage of fully ripe (BBCH 89). The winter wheat was harvested by the harvester Sampo Rosenlew 2035 (Sampo Rosenlew Ltd., Pori, Finland).

**Table 4.** Terms of sowing, fertilization, and harvest.

| Growing Season (GS) | Sowing | T1 (BBCH 25) | T2 (BBCH 32) | T3 (BBCH 50) | Harvest (BBCH 89) |
|---|---|---|---|---|---|
| GS1: 2017–2018 | 6 October 2017 | 5 March 2018 | 9 April 2018 | 2 May 2018 | 4 July 2018 |
| GS2: 2018–2019 | 9 October 2018 | 28 February 2019 | 29 April 2019 | 10 May 2019 | 11 July 2019 |
| GS3: 2020–2021 | 8 October 2020 | 3 March 2021 | 20 April 2021 | 29 May 2021 | 24 July 2021 |

BBCH−phase of growing according to Lancashire [44], BBCH 25–tillering; BBCH 32−stem elongation, BBCH 50–beginning of heading, BBCH 89–fully ripe.

The distribution of precipitation was not regular during the experimental years, as presented in Figure 1. The highest amount of precipitation was noticed in the GS3 (total precipitation of 377.20 mm/GS3). The GS1 and GS2 both had less rainfall and almost the same amount of precipitation (the total precipitation of 286.06 mm/GS1 and 282.40 mm/GS2). In the GS2, the lowest temperature was recorded in January while the other terms had their minimum temperatures in February. The GS2 and GS3 showed a similar temperature and precipitation development in the periods of May and June. The May period was colder and rainier while the temperature sharply rose in June. May in the GS3 was a little bit richer in precipitation than May in the GS2.

### 2.2. Yield and Grain Quality Measurement

The parameters observed over all experimental years were the grain yield and qualitative parameters such as the hectoliter weight of the grain, the content of protein and gluten in the grain, and the Zeleny-test (ZT) value. Four repetitions from all variants were analyzed. The weight of the harvested grain was determined using the digital scale Kern ECE 20K-2N (KERN and Sohn GmbH, Balingen, Germany). The test weight scale Wile 241 (Farmcomp OY, Tuusula, Finland) was used for determination of the hectoliter weight. The content of protein in the grain was determined by the Kjeltec 2300 device (Foss, Hillerød, Denmark) followed by the multiplication by a 6.25 coefficient (the Kjeldal method). The gluten content and the Zeleny-test value were estimated by the NIR (Near Infrared Spectroscopy) method on the Inframatic 9500 NIR grain analyzer (Perten Instruments, Hägersten, Sweden). The principle of the NIR method is the transmittance or reflectance measurement of radiation within the wavelength range of 800 to 2500 nm (12,500–4000 $cm^{-1}$) which is related to the different chemical groups contained in the sample [45,46].

### 2.3. Statistical Analysis

The program Statistica 12 CZ [47] was used for the statistical evaluation of monitored parameters. The Shapiro–Wilk and the Levene tests (at $p \leq 0.05$) were performed for the verification of normality and homogeneity of variances. The values of these parameters were subsequently evaluated by the analysis of variance (ANOVA) and by the follow-up tests according to Fisher (LSD test) at the 95% level ($p \leq 0.05$) of significance. The results were expressed as the arithmetical mean $\pm$ standard error (SE).

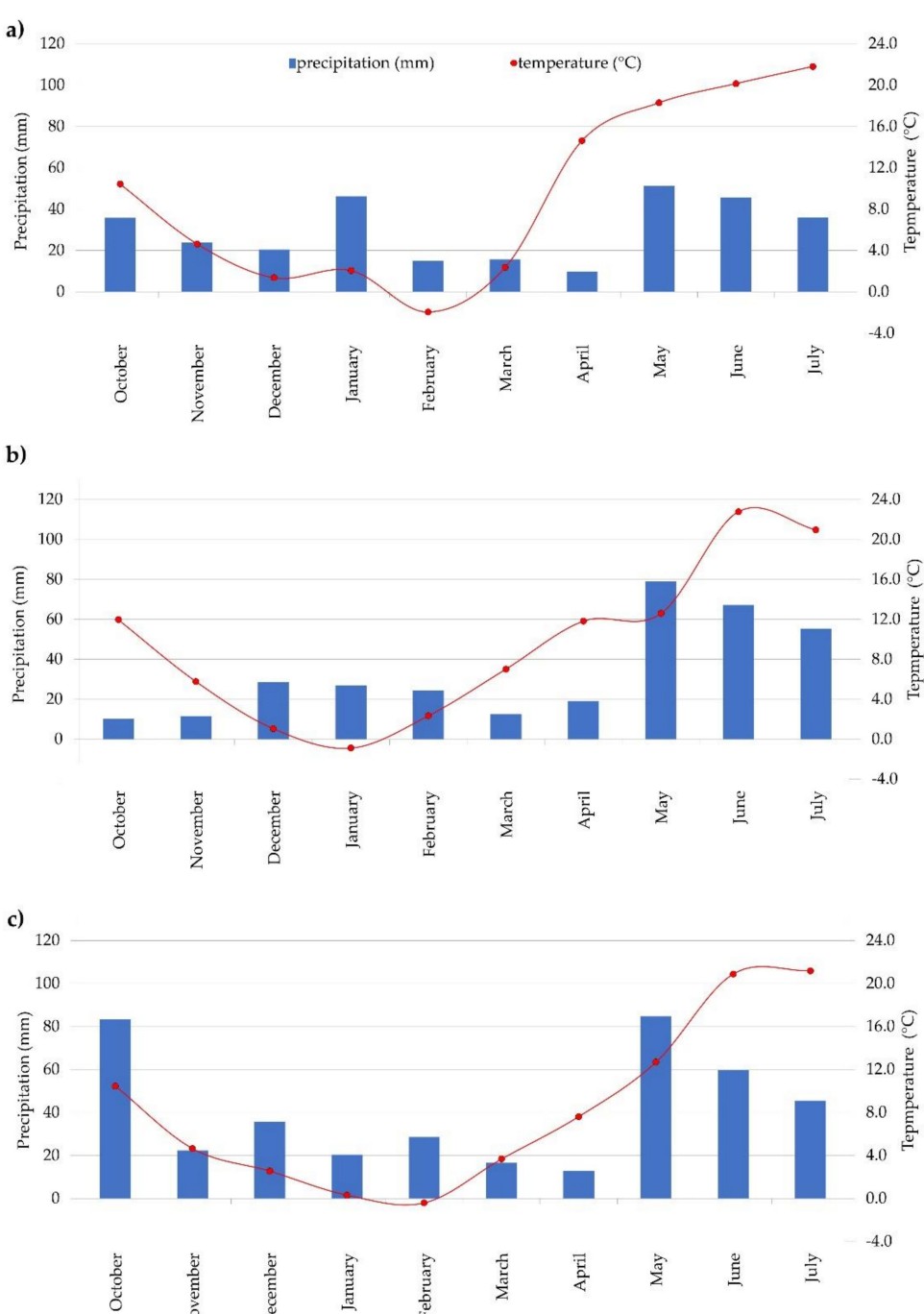

**Figure 1.** Weather conditions of the field experiments. (**a**) growing season 1 (2017−2018), (**b**) growing season 2 (2018−2019), (**c**) growing season 3 (2020−2021).

## 3. Results

### 3.1. The Grain Yield

The average grain yields over three growing seasons as well as the average grain yields in each year of the experiment are given in Figure 2. Almost no differences between the grain yield in the GS1 and GS2 were found. The yield of the N1−N4 treatments and the NS1−NS4 treatments were slightly increased in comparison to the control treatment, but the differences were statistically insignificant. The statistical differences were observed in the GS3. The significantly highest yield was observed in the N1 treatment fertilized by ALZON neo-N (urea with NI and UI), which was 6.3% higher than the control without

the application of inhibitors. The N1 treatment was also significantly higher than the NS1 treatment in the GS3.

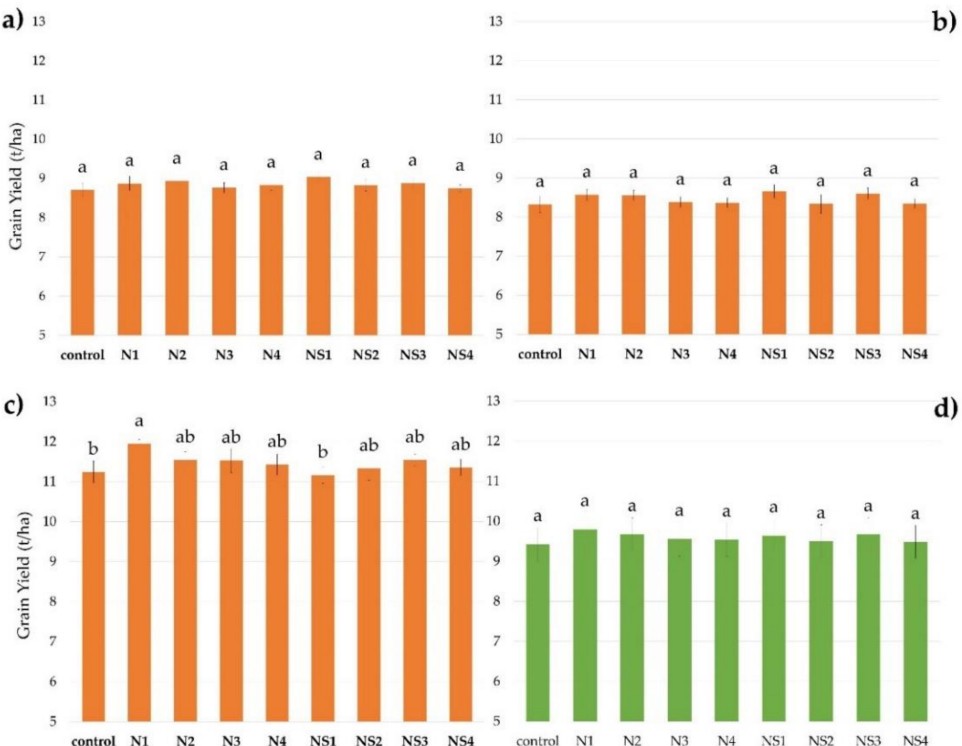

**Figure 2.** The effect of fertilizer treatments on grain yield (t/ha), (**a**) growing season 1 (2017−2018), (**b**) growing season 2 (2018−2019), (**c**) growing season 3 (2020−2021), (**d**) average yield. The same letters above the columns describe no statistically significant differences between treatments (LSD test). Each growing season was evaluated separately. Standard error (SE) is expressed by error bars.

According to the average values, the split fertilization did not result in an increased yield or a yield reduction. The treatments with the single application of N fertilizers with inhibitors provided a slightly increased yield in comparison to the treatments with the split application of N. The combination of nitrogen and sulfur fertilization showed a similar trend with the average increase of 2.1% in the NS3 treatment (the single application) in comparison with the NS4 (the split application).

## 3.2. Qualitative Parameters of Wheat

According to the standard of the Commission Regulation (EC) No. 824/2000, the required minimum value of the hectoliter weight of the wheat grain is 73 kg/hL. All of the treatments complied with this standard. The differences among the hectoliter weights of the individual fertilizer treatments were minimal in the GS1 and the GS2. Significant differences were observed in the GS3 (Table 5). On average, the hectoliter weight on the N2 treatment (the split application of CAN and ALZON neo-N) was significantly higher compared to the NS3 and NS4 treatments (with the sulfur application).

**Table 5.** The effect of fertilizer treatments on the hectoliter weight of the wheat grain (kg/hL).

| Treatments | GS1 | GS2 | GS3 | Average |
|:---:|:---:|:---:|:---:|:---:|
| control | 80.43 [a] ± 0.09 | 80.85 [a] ± 0.26 | 80.50 [abc] ± 0.11 | 80.59 [ab] ± 0.11 |
| N1 | 80.68 [a] ± 0.22 | 80.93 [a] ± 0.22 | 80.58 [ab] ± 0.23 | 80.73 [ab] ± 0.12 |
| N2 | 80.98 [a] ± 0.37 | 81.03 [a] ± 0.45 | 80.83 [a] ± 0.15 | 80.94 [a] ± 0.18 |
| N3 | 80.33 [a] ± 0.49 | 81.20 [a] ± 0.07 | 80.20 [bc] ± 0.07 | 80.58 [ab] ± 0.20 |
| N4 | 80.73 [a] ± 0.18 | 80.95 [a] ± 0.19 | 80.50 [abc] ± 0.22 | 80.73 [ab] ± 0.12 |
| NS1 | 80.88 [a] ± 0.11 | 81.13 [a] ± 0.16 | 80.30 [bc] ± 0.15 | 80.77 [ab] ± 0.13 |
| NS2 | 80.43 [a] ± 0.19 | 80.95 [a] ± 0.27 | 80.33 [bc] ± 0.21 | 80.57 [ab] ± 0.14 |
| NS3 | 80.45 [a] ± 0.17 | 81.05 [a] ± 0.31 | 80.08 [c] ± 0.10 | 80.53 [b] ± 0.16 |
| NS4 | 80.60 [a] ± 0.16 | 80.65 [a] ± 0.33 | 80.23 [bc] ± 0.17 | 80.59 [b] ± 0.13 |

Results are expressed as the mean ± standard error. The mean values with different letters are significantly different ($p < 0.05$) according to the LSD test (each growing season was evaluated separately). GS1—growing season 1 (2017–2018), GS2—growing season 2 (2018−2019), GS3—growing season 3 (2020−2021).

The results presented in Figure 3 clearly demonstrate an inconsistent protein content in the individual growing seasons. All treatments accomplished the minimal 10.5% value of the protein content set by the standard of the Commission Regulation (EC) No. 824/2000. The protein contents from the GS2 were higher compared to other growing seasons. Further, they were very uniform with no significant differences among the observed fertilization treatments. The protein contents observed in the GS1 and the GS3 showed more inconsistent values. In the GS1, the protein content in the N1 treatment was significantly lower compared with the N4 and NS2 treatments. These treatments were also significantly increased compared to the control; both were higher by 1.8%. The opposite trend was observed in the GS3. The protein content in the wheat grain of the N1 treatment was higher compared with the N4 and NS2 treatments. A significant difference was observed only in comparison with the NS2 treatment, which was lower by than N1 10.5%. The N1 treatment was distinguished by the highest protein content in the GS3 and in the average values. The protein content of the N1 treatment was higher by 5.4% than the control in the GS3 and higher by 2.3% than the control in the total average values.

The single application (N1, N3, and NS3 treatments) of fertilizers proved to have a significant effect on the protein content, which was increased in comparison with the split application (N2, N4, NS1, NS2, and NS4 treatments) in the GS3.

The gluten content is not commonly used in the EU countries as a technological quality criterion for the wheat grain exported to the food industry. Nevertheless, the gluten content is an important indicator of baking quality, which influences the properties of dough and bakery products. It is obvious from Figure 4 that the N2, N4, NS1, NS2, and NS4 treatments had a significantly higher gluten content compared to the control treatment in the GS1. The highest increase in the gluten content was found in the N4 treatment (higher by 3.4% than the control). The GS2 did not induce any significant differences in the gluten content among the treatments. Such results are in contrast with the GS3, which showed a slight decrease in the gluten content in comparison with the other terms. A significant difference was observed between the N1 treatment (the single application) and the N4, NS2, and NS4 treatments (with two applications). The N4 treatment was lower by 9.5%, the NS2 treatment by 10.1% and the NS4 treatment by 8.9% than the N1 treatment. The average values indicated the enhancement in the N1 (by 3.6%), N2 (by 2.5%), NS1 (by 1.9%), and NS3 treatments (by 1.8%) compared to the control.

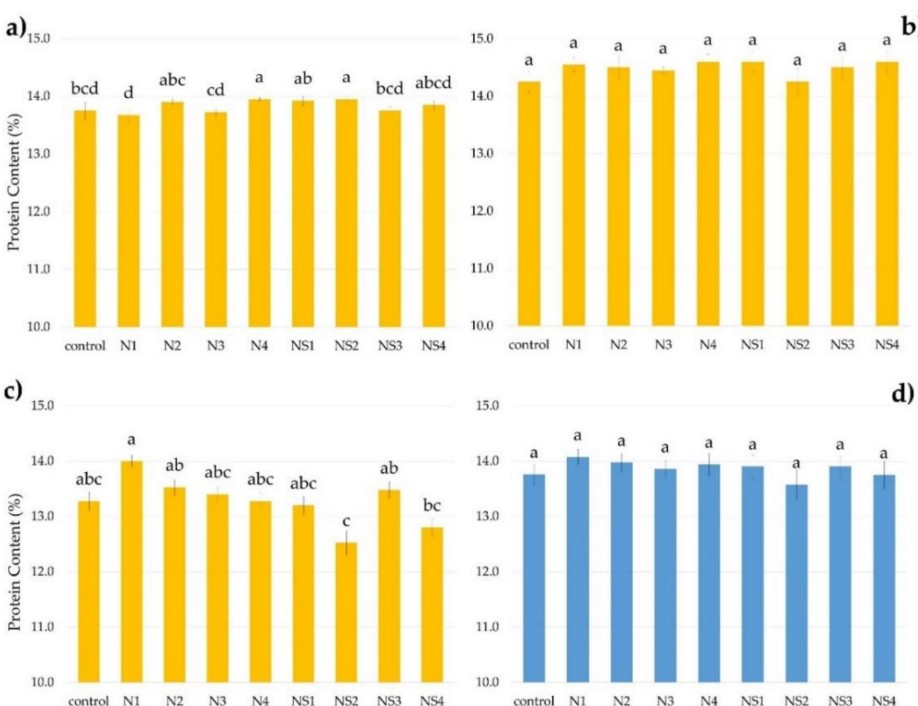

**Figure 3.** The effect of fertilizer treatments on protein content (%) in the grain, (**a**) growing season 1 (2017–2018), (**b**) growing season 2 (2018–2019), (**c**) growing season 3 (2020–2021), (**d**) average yield. The same letters above the columns describe no statistically significant differences between treatments (LSD test). Each growing season was evaluated separately. Standard error (SE) is expressed by error bars.

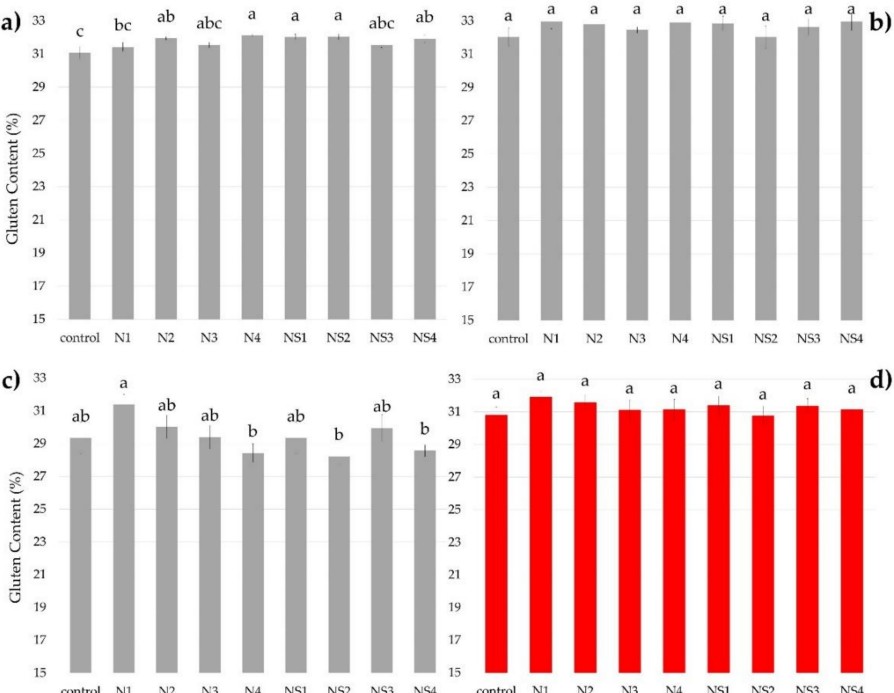

**Figure 4.** The effect of fertilizer treatments on gluten content (%) in the grain, (**a**) growing season 1 (2017–2018), (**b**) growing season 2 (2018–2019), (**c**) growing season 3 (2020–2021), (**d**) average yield. The same letters above the columns describe no statistically significant differences between treatments (LSD test). Each growing season was evaluated separately. Standard error (SE) is expressed by error bars.

The gluten content was significantly influenced by the application of two doses of fertilizer in the GS1 when it was higher than the gluten content in the control treatment (the application of three doses). A significant increase in the gluten content was observed in the treatments with the single application compared to the treatments with the split fertilization in the GS3 (two and three applications).

The values of the Zeleny test, which are displayed in Table 6, showed significant differences between the treatments in the GS1 and the GS3. The minimal value of ZT is 22 mL, according to the standard of the Commission Regulation (EC) No. 824/2000, and all the determined values were above this standard. Significant differences were observed in the N2, N4, NS1, and NS2 treatments in comparison with the control treatment in the GS1. The treatments without the sulfur fertilization (N2 and N4) were increased by 8%, the treatments fertilized with S (NS1 and NS2) showed ZT values that were higher by 7.3% compared to the control in the GS1. The highest ZT value was found in the N1 treatment in the GS3, which was significantly higher than the control (by 16.5%). In the N3, N4, and NS2 treatments, the ZT values were even higher by 17.1%, and by 14.1% in the NS4 treatment. A significant difference was also found between the NS3 (the single application) and the NS4 treatments (two applications of fertilizer) whereas the NS3 had a higher ZT value by 8.9%.

**Table 6.** The effect of fertilizer treatments on the Zeleny-test values (mL).

| Treatments | GS1 | GS2 | GS3 | Average |
|---|---|---|---|---|
| control | 37.8 [c] ± 0.3 | 44.0 [a] ± 1.8 | 44.1 [de] ± 1.6 | 41.9 [b] ± 1.2 |
| N1 | 39.0 [bc] ± 0.6 | 47.0 [a] ± 1.2 | 51.3 [a] ± 0.6 | 45.8 [a] ± 1.6 |
| N2 | 40.8 [a] ± 0.3 | 46.5 [a] ± 1.7 | 50.0 [ab] ± 0.7 | 45.8 [a] ± 1.3 |
| N3 | 39.0 [bc] ± 0.4 | 45.8 [a] ± 1.0 | 47.3 [bcd] ± 1.3 | 44.0 [ab] ± 1.2 |
| N4 | 40.8 [a] ± 0.3 | 46.8 [a] ± 1.3 | 47.7 [bc] ± 1.3 | 45.1 [ab] ± 1.1 |
| NS1 | 40.5 [a] ± 0.5 | 46.5 [a] ± 1.3 | 48.1 [abc] ± 0.9 | 45.0 [ab] ± 1.1 |
| NS2 | 40.5 [a] ± 0.5 | 44.0 [a] ± 2.2 | 43.9 [e] ± 0.5 | 42.8 [ab] ± 0.9 |
| NS3 | 39.0 [bc] ± 0.4 | 45.8 [a] ± 1.7 | 49.0 [ab] ± 1.9 | 44.6 [ab] ± 1.5 |
| NS4 | 40.0 [ab] ± 0.7 | 47.0 [a] ± 1.8 | 45.0 [cde] ± 0.7 | 44.0 [ab] ± 1.0 |

Results are expressed as the mean ± standard error. The mean values with different letters are significantly different ($p < 0.05$) according to the LSD test (each growing season was evaluated separately). GS1−growing season 1 (2017−2018), GS2−growing season 2 (2018−2019), GS3−growing season 3 (2020−2021).

Significant differences were observed between each type of fertilization (single, split and control) in the GS1. The highest values of ZT were observed in the treatments with the split application of fertilizers. Differences between effects of three and one or two applications on the ZT values were significant in the GS3. The average values showed a significant difference between the control (three applications of fertilizers) and the treatments with one application (N1, N3, and NS3).

## 4. Discussions

The application of fertilizers with the NIs leads to partial ammonium nutrition in plants which could affect the crop yield and quality. Many studies have observed this effect, but the results diverge. Some studies have not found any changes in the wheat yield after the use of NI [48–50]. On the other hand, other studies describe a slight increase in the wheat yield after the NI application [51,52]. Our examined treatment that was fertilized by ammonium sulphate nitrate (ASN) with the NIs of dicyandiamide (DCD) in combination with triazole (TZ) (the NS3 treatment, fertilizer ENSIN) did not provide any significant increase in the grain yield, although the yield was slightly higher compared to the control treatment. A significant increase in the grain yield was observed after the fertilization with ALZON neo-N (the N1 treatment, urea with NI and UI) compared to the control and the NS1 treatment in the moisture-certain GS3. Other studies in the literature [53–55] report

that many environmental factors including rainfalls and soil moisture have an effect on NIs as reducers of $N_2O$ emissions, which subsequently influences the amount of N available to plants. The increase in the wheat yield after the split application of conventional N fertilizers was described in [16], which is contrary to our results in the N1 treatment with the single application in the GS3. A relative yield increase in the N1 treatment was observed in the GS1 and the GS2. The other treatments with the single application of fertilizers with NI and/or UI (N3 and NS3) also showed a relative increase in the yield compared to the treatments with the split application in each growing season. The simplification of the split application from two or three doses into a single application is an evident benefit of the use of NI and UI [56].

Grain quality is assessed by physical and chemical characteristics, which are influenced by genetic potential and environmental conditions [57]. The hectoliter weight of the wheat grain is a parameter of milling quality, and it is also connected with the wheat milling yield. It depends on agricultural inputs, variety, and weather conditions [58]. The average hectoliter weights of the wheat grain were slightly above 80.4 kg/hL in our experiment, which was the value declared by the plant breeder [43]. According to the average values, the S application did not result in an increase in this parameter. In the N2 treatment (without sulfur application), the value was significantly higher than in the NS2 and NS4 treatments (with the sulfur application). This result is similar to that reported by Hoel [59], who found a significant decrease in the hectoliter weight of the wheat grain after the S fertilization.

A study by Abalos et al. [60] explained that the application of NIs could positively affect the grain quality due to the inhibitors' induction followed by a subsequent increase in the $NH_4^+$ nutrition and a decrease in the N losses by the leaching of $N_2O$ emissions. Some studies [48,51,61] have not found any changes in the grain quality after different NI applications, while Guardia et al. [62] observed slight differences after the application of the DMPSA (3,4-Dimethylpyrazole-succinic acid) inhibitor. Regarding the protein content, the N1 treatment (urea with NI and UI) showed a relative enhancement of this parameter. The increase in the grain protein content after the fertilization by the ammonium-nitrate fertilizer with NI dicyandiamide (DCD) was reported by Peltonen and Virtanen [63]. Our results are in contrast with this study because a relatively slight increase in the grain protein content was observed in the NS3 and NS4 treatments with DCD and TZ inhibitors (fertilizer ENSIN). The references on the impact of the sulfur fertilization on the grain protein content are inconsistent. Some studies [64–66] described no effect of S on the protein content, while another [67] reported the increase in grain protein after the S application. A significant increase in the protein content was observed in the NS2 treatment compared with the control and the N1, N3, NS3, and NS4 treatments, but the average values showed neither a decrease nor an increase in the protein content after the S fertilization. Järvan et al. [68] claimed that weather conditions as well as the sulfur application affect whether the correlation between the grain yield and the protein content is positive or negative. Other studies [69,70] suggested that the grain protein content is influenced by the late N fertilization. The significant effect of the split application on the increased protein content in the grain was observed in the N4 and NS2 treatments (the split application) compared to the treatments with the single application in the GS1 (the season with the lower level of precipitation). On the contrary, a stronger impact of the single application on the level of grain protein was observed in the more moisture-certain GS3.

The gluten content in the grain is affected by the amount of N supplied [71]. When comparing the gluten content of the N1 treatment (urea with MPA and 2-NPT inhibitors) and the NS4 treatment (ASN with DCD and TZ inhibitors) in the rainfall-rich GS3, we noticed a significantly higher gluten content in the N1 treatment. The studies conducted by Cahalan et al. [72] and by Shepherd et al. [73] found a correlation between the increased levels of precipitation and the higher leaching of the DCD inhibitor. The leaching of DCD can cause a lower amount of available N as well as a subsequent decrease in the gluten content in the GS3, in contrast to the drier GS1 and GS2. McGeough et al. [74] even stated

that DCD showed a higher inhibition effect on arable soils. It is known that sulfur plays an important role in the amount of gluten [75] and in the composition of gluten proteins [65,76]. Nevertheless, the positive effect of the S fertilization on the gluten content was observed only in the GS1, when the NS1, NS3, and NS4 treatments provided a significantly higher content of gluten than the control. The significant impact of the split N application on the increased gluten content was found in the N2, N4, NS1, NS2, and NS4 treatments in the GS1, which is consistent with studies [18,77] describing the increase in gluten concentration after the split N application. On the contrary, a significant difference was observed between the N1 treatment with the single application and the treatments with two applications (N4, NS2, and NS4) in the GS3.

The Zeleny test represents viscoelastic characteristics of gluten proteins, which determine baking quality [78]. Meteorological factors such as temperature and rainfall during the wheat growth strongly influence this parameter [79]. The significantly highest values of the ZT in the GS3 as well as on average were observed in the N1 treatment (compared to the control). Further, the application of fertilizers with NI and UI positively influenced this quality parameter under the conditions with higher precipitation. Although other studies [66,80] reported that the S application could positively influence the value of ZT, the effect of S fertilization on the ZT values was variable in this experiment. The NS1 and NS2 treatments (fertilized with N and S) had significantly increased ZT values compared to the control in the GS1. Nevertheless, the same effect of S was not found in the other growing seasons. The single application of fertilizers brought a significant impact on the ZT, whereas the values in these treatments (N1, N3, and NS3) were higher in comparison with the control treatment (three doses of fertilizer).

Urease activity is dependent on soil moisture because the rate of urea hydrolysis is low in dry conditions [81]. The response of wheat to the inhibited fertilizer UREA[stabil] (the N3 treatment) was influenced by weather conditions in this experiment. The relatively long period without rainfall, which was recorded after the first fertilization date (T1) in the seasons GS1 (12 days) and GS2 (11 days), increased the effectiveness of the UI. Nitrogen contained in this fertilizer (the N3 treatment) was largely retained in the soil for later use, which had a positive effect on the grain quality (protein and gluten content). The GS3 was characterized by different rain distribution. A rainfall of 12.4 mm occurred the second day after the first term of the N application (T1), and the values of the protein and gluten content in the N3 treatment (UREA[stabil]) were lower in comparison with the drier conditions of the GS1 and the GS2. The conditions for the UI activity were not suitable in the GS3 because the rain could mitigate the gaseous loss. It could also increase the risk of leaching, which probably occurred in this case. Subsequently, the rain negatively influenced the protein and gluten content in this treatment. A highly critical factor in the application of inhibitors is the timing of their application. Their effect on the grain yield and the N efficiency is strongly affected by environmental conditions during the period of their probable inhibition activity (a period of 1−8 weeks) [82]. Chien et al. [56] stated that it is convenient if a rainfall comes 5−7 days after the fertilization by urea with inhibitors, i.e., at the time of a potentially high inhibitor activity. It is also reported that UI is suitable for arable soils on which they are capable of reducing $NH_3$ emission, and thus they create a better opportunity for plants to use N at later growth phases [83,84]. In contrast, the treatments with ALZON neo-N fertilizer containing both types of inhibitors showed higher values of quality parameters in the rainfall-rich GS3 compared to the treatments fertilized with the fertilizer containing only UI. These results indicate a greater effect of fertilization with NI and UI on the better N availability for plants in more humid areas.

## 5. Conclusions

The application of N fertilizers with NI and/or UI has great potential in today's agriculture. The cultivation of winter wheat requires high inputs of N fertilization and the application of inhibited fertilizers can reduce the risk of N losses, which negatively affect the environment as well as the efficiency of N fertilization. Neither the grain yield

nor the wheat quality was reduced after the single application of fertilizers with inhibitors. In addition, a relatively average increase in the observed parameters was noticed. The protein content and ZT value were even significantly higher after the single application in comparison with the split application. Differences in the effects of the applied fertilizers with inhibitors on wheat in relation to the rainfall in each growing season were observed. The fertilizer with NI and UI ALZON neo-N (the N1 treatment) seemed to be more effective in the moisture-rich conditions of the GS3 due to the significant increase in the grain yield and ZT value in the GS3 were observed, as well as to the other observed parameters that were also relatively increased in contrast to the application under drier conditions. The use of fertilizers containing N-transformation inhibitors in wheat nutrition provides the benefit of the possibility of combining split N rates. Reducing the number of land entries has an impact on the economics of wheat cultivation, reduces soil compaction and contributes to environmental protection.

**Author Contributions:** Conceptualization, M.Š. and P.Š.; methodology, M.Š. and P.Š.; validation, M.Š., P.Š. and J.A.; formal analysis, M.Š. and P.Š.; investigation, M.Š., P.Š. and J.A.; resources, M.Š.; data curation, P.Š; writing—original draft preparation, M.Š.; writing—review and editing, P.Š., P.R., Z.K. and J.A.; visualization, M.Š. and J.A.; supervision, P.Š.; project administration, M.Š., P.Š. and J.A. All authors have read and agreed to the published version of the manuscript.

**Funding:** This research received no external funding.

**Institutional Review Board Statement:** Not applicable.

**Informed Consent Statement:** Not applicable.

**Data Availability Statement:** Presented data in this study are available on request from the corresponding author.

**Conflicts of Interest:** The authors declare no conflict of interest.

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
