# Peer review of "Response of Winter Wheat (Triticum aestivum L.) to Fertilizers with Nitrogen-Transformation Inhibitors and Timing of Their Application under Field Conditions"

_agronomy, doi:10.3390/agronomy12010223_

Round 1

Reviewer 1 Report

This manuscript compares the effectiveness of conventional N fertilizer application Vs. N fertilizer with inhibitors in winter wheat. This study is helpful to improve the understanding of nitrogen response in winter wheat. There are several typos in the manuscript. Please pay attention to these errors. A major issue of this study is that past studies have shown that crops get nitrogen from applied and carryover nitrogen from the previous year, and the accumulation of carryover N significantly affects crop yield and quality. Without accounting for carryover, residual nitrogen in crop production may not be efficient, and recommended nitrogen levels may be sub-optimal (under or over). Authors should acknowledge it. Please see the following literature. 

Raun, W. R., Dhillon, J., Aula, L., Eickhoff, E., Weymeyer, G., Figueirdeo, B., ... & Fornah, A. (2019). Unpredictable nature of environment on nitrogen supply and demand. Agronomy Journal111(6), 2786-2791.

Line 12: "UI" after inhibitors.

Line 66: Please check the reference.

line 147: what is Statistica 12 CZ?

Be precise on the amount of nitrogen you recommend based on this study.

Please cite and provide references in accordance with the journal's format.

Please check the text format throughout the text and maintain consistency. 

Author Response

Dear reviewer,

I am sending a revised and corrected manuscript “Response of winter wheat (Triticum aestivum L.) to fertilizers with nitrogen transformation inhibitors and timing of their application under field conditions " where we accepted most of the comments and suggestions.

Thank you for your very careful review of our paper, and for the comments, corrections and suggestions. Revisions to the paper have been made taking most of the above into account. We believe that this has significantly improved our manuscript. Below are the specific answers to the comments:

This manuscript compares the effectiveness of conventional N fertilizer application Vs. N fertilizer with inhibitors in winter wheat. This study is helpful to improve the understanding of nitrogen response in winter wheat. There are several typos in the manuscript. Please pay attention to these errors. A major issue of this study is that past studies have shown that crops get nitrogen from applied and carryover nitrogen from the previous year, and the accumulation of carryover N significantly affects crop yield and quality. Without accounting for carryover, residual nitrogen in crop production may not be efficient, and recommended nitrogen levels may be sub-optimal (under or over). Authors should acknowledge it. Please see the following literature. 

Raun, W. R., Dhillon, J., Aula, L., Eickhoff, E., Weymeyer, G., Figueirdeo, B., ... & Fornah, A. (2019). Unpredictable nature of environment on nitrogen supply and demand. Agronomy Journal111(6), 2786-2791.

  • We are aware that plant growth is strongly influenced by soil residual N supply, which is highly variable depending on soil and climatic conditions. Soil N content was determined, not only before sowing but also during the growing season. The effect of fertilizer application on soil mineral N supply was also evaluated. This information is not the subject of the present study. However, we plan to publish the results with reference to the current manuscript. In our study, we try to interpret the results of the field experiments more in terms of the agronomic impact of fertilization on grain production and quality. We are aware that N release from fertilizers, which affects soil residual N supply, is significantly related to their effect on plant production. Nevertheless, we would like to apply the results describing the effect of fertilizer application with inhibitors on soil nitrogen content in journals that focus on soil or environmental nutrient regimes.

Line 12: "UI" after inhibitors.

  • Thank you for your comment. The sentence contained a typo and has been corrected.

Line 66: Please check the reference.

  • Thank you for your comment. The sentence contained a typo and has been corrected.

Line 147: what is Statistica 12 CZ?

  • Statistka 12 Cz is a statistical data processing program that was used to evaluate the measured parameters.

Be precise on the amount of nitrogen you recommend based on this study.

  • The present study compares the effect of conventional N and NS fertilizers with fertilizers containing N conversion inhibitors. The N rate was the same for all the variants studied (160 kg/ha N). The advantage of using them (at the same dose that was tested) appears to be the possibility of combining individual fertiliser applications into a single dose while maintaining the same effect as a split application. Reducing the number of land entries has an impact on the economics of wheat cultivation, reduces soil compaction and contributes to environmental protection.

Please cite and provide references in accordance with the journal's format.

  • Thank you for your comment. The formatting of the references has been modified according to the Instructions for Authors.

Please check the text format throughout the text and maintain consistency. 

  • Thank you for your comment. The manuscript was edited for language and was checked and edited according to the template in the Instructions for Authors.

Reviewer 2 Report

This manuscript presents interesting results of the effect of fertilizers with nitrogen transformation inhibitors and timing of their application on winter wheat yield and grain quality under field conditions. The adopted approach in this study was very interesting since it could reduce N losses and thereby improving plant performances as well as reducing fertilizers negative impact on the environment.

The manuscript was well introduced, and the authors adopted very convincing methods with a consistent discussion of the different obtained results. However, the manuscript needs some revisions, particularly English review.

General comments

- Comment 1: The English of this manuscript needs substantial improvements.

- Comment 2: The reference citations format to correct in the whole manuscript.

- Comment 3: You have to put a dot (not a coma) at the end of the sentence and start a new one (check the whole manuscript).

Comment 4: It is not clearly mentioned in M&M section, did you perform the experimentation in the same plots each year or in new plots every year?

Comment 5: It would be interesting if you measured soil mineral content to have more precise idea on the effect of NI and UI since we can’t extrapolate on plant without seeing what happened in the soil.

Other comments

- Title: please change in field to under field

- Abstract

L12: please delete “how”

L13-15: please rewrite

L23: please add wheat to the keywords

- Introduction

L35: please delete “have an impact on”

L47: please delete “how”

L56: please change “Nis” to “Nis”

L55-58: please rewrite

L64: please delete “how”

L64: please change “to minimalize negative” to “to minimalize the negative”

L65: please channge “of fertilization, their positive” to “of fertilization since their positive”

L67: please change “right” to “high”

L67: please change “sulphur (S), it plays” to “sulphur (S) which plays”

L76: please change “yield and quality” to “grain yield and quality”

- M&M

L100: please delete “legend”

L107: please delete “legend”

L117: please delete “legend”

- Results

L161: please delete “by”

L183: Table 4: the use of the letter of signification should be the same in figures and tables; in figure 2 you assigned “a” to the maximum value and the opposite in table 4. It is the same in the other tables. Please correct

L184: please delete “legend”

L196: please change “also significant increased” to “also significantly increased”

L253: please delete “legend”

- Discussion

L285: please change “mentioned Hoel [51] who found significant decreased” to “mentioned by Hoel [51] who found a significant decrease”

L293: please change “enhance of this parameter. Increase” to “enhancement of this parameter. The increase”

L298: please change “different” to “inconsistent”

L334: please channge “Treatments NS1 and NS2” to “NS1 and NS2 treatments”, the same thing in the whole manuscript

- Conclusion

L369: please change “the relative average increase in observed parameters was determined” to “a relative average increase in the observed parameters was noticed”.

Author Response

Dear reviewer,

I am sending a revised and corrected manuscript “Response of winter wheat (Triticum aestivum L.) to fertilizers with nitrogen transformation inhibitors and timing of their application under field conditions " where we accepted most of the comments and suggestions.

Thank you for your very careful review of our paper, and for the comments, corrections and suggestions. Revisions to the paper have been made taking most of the above into account. We believe that this has significantly improved our manuscript. Below are the specific answers to the comments:

This manuscript presents interesting results of the effect of fertilizers with nitrogen transformation inhibitors and timing of their application on winter wheat yield and grain quality under field conditions. The adopted approach in this study was very interesting since it could reduce N losses and thereby improving plant performances as well as reducing fertilizers negative impact on the environment.

The manuscript was well introduced, and the authors adopted very convincing methods with a consistent discussion of the different obtained results. However, the manuscript needs some revisions, particularly English review.

General comments

Comment 1: The English of this manuscript needs substantial improvements.

and

Comment 3: You have to put a dot (not a coma) at the end of the sentence and start a new one (check the whole manuscript).

  • The English of the manuscript has been revised. We believe that the English language and style are fine.

Comment 2: The reference citations format to correct in the whole manuscript.

  • Thank you for your comment. The formatting of the references has been modified according to the Instructions for Authors.

Comment 4: It is not clearly mentioned in M&M section, did you perform the experimentation in the same plots each year or in new plots every year?

  • Thank you for your comment. The text has been added as recommended.

Comment 5: It would be interesting if you measured soil mineral content to have more precise idea on the effect of NI and UI since we can’t extrapolate on plant without seeing what happened in the soil.

  • The effect of fertilization on soil and plant nitrogen content was evaluated. This information is not the subject of the present study. However, we plan to publish the results with reference to the current manuscript. In our study, we try to interpret the results of the field experiments more in terms of the agronomic impact of fertilization on grain production and quality. We are aware that N release from fertilizers, which affects soil residual N supply, is significantly related to their effect on plant production. Nevertheless, we would like to apply the results describing the effect of fertilizer application with inhibitors on soil nitrogen content in journals that focus on soil or environmental nutrient regimes.

Other comments

Title: please change in field to under field

  • Thank you for your comment. The title has been corrected.

Abstract

L12: please delete “how”

L13-15: please rewrite

L23: please add wheat to the keywords

  • Thank you for your comment. The abstract has been corrected as recommended.

Introduction

L35: please delete “have an impact on”

L47: please delete “how”

L56: please change “Nis” to “Nis”

L55-58: please rewrite

L64: please delete “how”

L64: please change “to minimalize negative” to “to minimalize the negative”

L65: please channge “of fertilization, their positive” to “of fertilization since their positive”

L67: please change “right” to “high”

L67: please change “sulphur (S), it plays” to “sulphur (S) which plays”

L76: please change “yield and quality” to “grain yield and quality”

  • Thank you for your comment. The “Introduction” has been corrected as recommended.

M&M

L100: please delete “legend”

L107: please delete “legend”

L117: please delete “legend”

  • Thank you for your comment. The “Materials and Methods” has been corrected as recommended.

Results

L161: please delete “by”

L183: Table 4: the use of the letter of signification should be the same in figures and tables; in figure 2 you assigned “a” to the maximum value and the opposite in table 4. It is the same in the other tables. Please correct

L184: please delete “legend”

L196: please change “also significant increased” to “also significantly increased”

L253: please delete “legend”

  • Thank you for your comment. The “Results” has been corrected as recommended.

Discussion

L285: please change “mentioned Hoel [51] who found significant decreased” to “mentioned by Hoel [51] who found a significant decrease”

L293: please change “enhance of this parameter. Increase” to “enhancement of this parameter. The increase”

L298: please change “different” to “inconsistent”

L334: please channge “Treatments NS1 and NS2” to “NS1 and NS2 treatments”, the same thing in the whole manuscript

  • Thank you for your comment. The “Discussion” has been corrected as recommended.

Conclusion

L369: please change “the relative average increase in observed parameters was determined” to “a relative average increase in the observed parameters was noticed”.

  • Thank you for your comment. The “Conclusion” has been corrected as recommended.

Round 2

Reviewer 1 Report

No further comments.

Author Response

Thank you for your very careful review of our paper, and for the comments, corrections and suggestions.
